# The Impact of Mindfulness on Functional Brain Connectivity and Peripheral Inflammation in Breast Cancer Survivors with Cognitive Complaints

**DOI:** 10.3390/cancers15143632

**Published:** 2023-07-15

**Authors:** Michelle Melis, Gwen Schroyen, Jeroen Blommaert, Nicolas Leenaerts, Ann Smeets, Katleen Van Der Gucht, Stefan Sunaert, Sabine Deprez

**Affiliations:** 1Department of Imaging and Pathology, Translational MRI, Catholic University Leuven, 1000 Brussels, Belgium; gwen.schroyen@hotmail.be (G.S.); stefan.sunaert@kuleuven.be (S.S.); sabine.deprez@kuleuven.be (S.D.); 2Research Foundation Flanders (FWO), 1000 Brussels, Belgium; 3Leuven Brain Institute, Catholic University Leuven, 3000 Leuven, Belgium; jeroen.blommaert@kuleuven.be (J.B.); nicolas.leenaerts@kuleuven.be (N.L.); 4Leuven Cancer Institute, Catholic University Leuven, 3000 Leuven, Belgium; ann.smeets@uzleuven.be; 5Department of Oncology, Gynecological Oncology, Catholic University Leuven, 3000 Leuven, Belgium; 6Department of Neurosciences, Mind-Body Research, Catholic University Leuven, 3000 Leuven, Belgium; 7Department of Oncology, Surgical Oncology, Catholic University Leuven, 3000 Leuven, Belgium; 8Department of Surgical Oncology, Multidisciplinary Breast Center, University Hospitals Leuven, 3000 Leuven, Belgium; 9Tilburg School of Social and Behavioral Sciences, Tilburg University, 5037 AB Tilburg, The Netherlands; katleen.vandergucht@kuleuven.be; 10Leuven Mindfulness Centre, Faculty of Psychology and Educational Sciences, Catholic University Leuven, 3000 Leuven, Belgium; 11Neuromodulation Laboratory, Biomedical Sciences Group, Department of Rehabilitation Sciences, Catholic University Leuven, 3000 Leuven, Belgium; 12Department of Radiology, University Hospitals Leuven, 3000 Leuven, Belgium

**Keywords:** breast cancer, mindfulness, cognition, MRI, functional connectivity, inflammation, graph theory, resting state

## Abstract

**Simple Summary:**

Cognitive impairment is a common side effect of cancer treatment and impacts the quality of life of cancer survivors. As there is currently no golden standard for the treatment of cancer-related cognitive impairment (CRCI), we investigated the potential of a mindfulness-based intervention to impact the underlying mechanisms of CRCI. Breast cancer survivors with cognitive complaints (*n* = 117) were randomly assigned to a mindfulness, physical training, or waitlist control group. Resting state functional MRI data and serum blood samples were collected and compared before and after the intervention. We could not identify differences between the groups in resting state functional connectivity. However, the functional organization of attention-, salience- and executive functioning-related neural networks differed between both intervention groups and the waitlist control group. Additionally, physical training could alter therapy-induced immune deregulation. In conclusion, physical training had the most pronounced effects on functional network organization and biomarkers of inflammation, two mechanisms that might be involved in CRCI.

**Abstract:**

Background: Cancer-related cognitive impairment (CRCI) has been linked to functional brain changes and inflammatory processes. Hence, interventions targeting these underlying mechanisms are needed. In this study, we investigated the effects of a mindfulness-based intervention on brain function and inflammatory profiles in breast cancer survivors with CRCI. Methods: Female breast cancer survivors reporting cognitive complaints (*n* = 117) were randomly assigned to a mindfulness-based intervention (*n* = 43), physical training (*n* = 36), or waitlist control condition (*n* = 38). Region-of-interest (ROI) and graph theory analyses of resting state functional MRI data were performed to study longitudinal group differences in functional connectivity and organization in the default mode, dorsal attention, salience, and frontoparietal network. Additionally, bead-based immunoassays were used to investigate the differences in inflammatory profiles on serum samples. Measures were collected before, immediately after and three months post-intervention. Results: No ROI-to-ROI functional connectivity changes were identified. Compared to no intervention, graph analysis showed a larger decrease in clustering coefficient after mindfulness and physical training. Additionally, a larger increase in global efficiency after physical training was identified. Furthermore, the physical training group showed a larger decrease in an inflammatory profile compared to no intervention (IL-12p70, IFN-γ, IL-1β, and IL-8). Conclusion: Both mindfulness and physical training induced changes in the functional organization of networks related to attention, emotion processing, and executive functioning. While both interventions reduced functional segregation, only physical training increased functional integration of the neural network. In conclusion, physical training had the most pronounced effects on functional network organization and biomarkers of inflammation, two mechanisms that might be involved in CRCI.

## 1. Introduction

Breast cancer is the most diagnosed cancer among women worldwide [1]. Earlier diagnosis and advancing treatment options have led to increased survival rates, which, in turn, increased the focus on quality of life [2]. Cancer treatment has been linked to various acute and long-term side effects, including effects on psychological well-being, fatigue, and cognitive performance [3]. More specifically, cancer-related cognitive impairment (CRCI) affects about 20–50% of breast cancer survivors after chemotherapy treatment, with impairments in attention, memory, executive functioning, and processing speed [4].

Changes in cognition after cancer treatment have been linked to widespread changes in both the structure and function of the brain [5]. For example, resting state functional MRI (rsfMRI) has been applied to assess functional connectivity between distant brain regions through spontaneous co-activation patterns [6]. Several studies found alterations in resting state functional connectivity in attention-, executive function-, and emotion-related networks of chemotherapy-treated patients compared to healthy controls. More specifically, changes in the default mode network (DMN) [7,8,9,10], dorsal attention network (DAN) [7,11], frontoparietal network (FPN) [10,12] and salience network (SN) [13] have been linked to CRCI. Additionally, functional connectivity changes in the hippocampus have been associated with CRCI. The hippocampus has been shown to be vulnerable to the effects of cancer treatment, and it plays a crucial role in memory processes and in the regulation of stress and emotions [14,15]. Additionally, the hippocampus has been considered a part of the DMN [16]. More recently, graph theory has been applied to rsfMRI data to study alterations in functional brain organization. Graph theory allows to study alterations in brain topology by defining nodes (brain regions) and edges (connectivity between the nodes), forming the so-called brain connectome. Hence, graph theory provides a tool to study organization and efficiency of the brain network, rather than focusing on individual brain areas, through metrics characterizing functional integration and segregation properties [17,18,19]. Integration refers to the ability to combine information from various distinct brain areas, whereas segregation refers to the specialization of brain areas [20]. Graph metrics of integration include characteristic path length and global efficiency, whereas segregation measures include clustering coefficient and local efficiency [21]. In breast cancer survivors, altered integration [22,23] and reduced segregation [20,22,24] have been reported compared to healthy controls.

Additionally, increased levels of pro-inflammatory markers have been associated with neural changes and decreased objective cognitive performance (e.g., memory and processing speed) and self-reported cognitive function after cancer treatment, although the latter evidence is still preliminary [25]. More specifically, chemotherapy might have indirect neurotoxic effects by increasing levels of pro-inflammatory cytokines (e.g., interleukin (IL)-6), and IL-1β) that can potentially cross the blood–brain barrier and evoke an inflammatory response in the brain, ultimately leading to cognitive impairment [26]. Moreover, one of the most robust biological markers of CRCI has been shown to be inflammatory (e.g., cytokine IL-6) [27]. Pro-inflammatory cytokines can be secreted in response to psychological factors like stress, or in response to physical factors like an immune response to chemotherapy, surgery or the tumor itself [28]. Increased levels of cytokines have been linked to neural changes like decreased hippocampal and insular volume might exert a direct effect on cognition [25]. Alternatively, cytokines might indirectly impact cognition by inducing sickness behavior like a depressed mood, fatigue, and social withdrawal [5].

Although more research is currently focusing on potential interventions for CRCI, there is still no golden standard for the treatment of CRCI [29,30]. In this study, which is part of a larger randomized controlled trial (RCT), we investigated the impact of a mindfulness-based intervention (MBI) on CRCI-related mechanisms compared to a physical training and waitlist control group. An MBI is an evidence-based intervention based on Buddhist tradition that combines group- and home-based exercises [31]. During MBIs, participants are invited to attend to internal and external experiences like bodily sensations, thoughts, emotions, or sounds with an attitude of openness and non-judgmental acceptance. MBIs have been shown to be cost-effective interventions for breast cancer patients and other chronic health conditions [32].

We hypothesized that MBIs might alleviate CRCI indirectly via several pathways. First, MBIs might alter functional connectivity between networks associated with CRCI in healthy and clinical populations, i.e., DMN, FPN, DAN, and SN [33]. In breast cancer survivors, resting state functional connectivity changes have been shown after MBI compared to a waitlist control group within the DMN [34] and between DAN and SN-related regions [35]. These findings were associated with reductions in pain severity [34] and emotional distress [35], respectively. However, both studies were limited by the lack of an active control condition and small sample sizes, emphasizing the need for larger studies with active control groups to verify these findings. Additionally, graph analysis might be a sensitive biomarker for functional reorganization after MBI, with decreases in functional segregation and increases in functional integration already shown in different non-cancer populations [36,37]. In healthy elderly, however, no pre- to post-MBI changes in whole-brain graph metrics could be identified [38]. Although the effects of cancer treatment on functional reorganization have previously been described in the literature, no studies have investigated the impact of MBI on functional network organization using graph theory in breast cancer survivors. Second, preliminary evidence showed that MBIs might reduce peripheral inflammation in various clinical and non-clinical populations. More specifically, levels of pro-inflammatory cytokines might be reduced in peripheral blood after MBIs, which in turn might decrease peripheral inflammation. However, the evidence for alterations in peripheral cytokine levels after MBIs remains mixed [39,40]. Similarly, in cancer populations, preliminary evidence showed that MBIs might alter peripheral cytokine levels, thereby reducing peripheral inflammation and associated neuronal damage [41]. However, large-scale randomized controlled trials with longer follow-up periods are needed to verify these findings. Furthermore, it has repeatedly been shown that MBIs can reduce symptoms of depression, anxiety, stress, and fatigue in breast cancer survivors [42,43] and various other populations [44], which might indirectly impact cognitive outcomes. In our study, emotional distress, fatigue, and quality of life improved after MBI and physical training but remained stable in the waitlist control group over time [45].

In this multimodal study, we investigated the impact of MBI on CRCI and related mechanisms compared to a physical training and a waitlist control group in breast cancer survivors reporting cognitive complaints. Physical training was used as an active control condition to control for non-specific intervention effects like being part of a group, interaction with the trainer, and expectations for improvement [46]. The physical training program was an extended and adapted version of the standard of care rehabilitation program at University Hospitals Leuven. Although the evidence for an effect of physical training on CRCI remains inconclusive [47,48], recent research showed promise for reducing subjective cognitive impairment in breast cancer survivors after physical training [49,50].

The first results of our study showed a decrease in subjective cognitive impairment in the three groups over time, without group differences. Additionally, no differences between the groups in objective cognitive impairment could be identified [45]. In this paper, we elaborate on the potential effects of MBI on resting state functional connectivity and functional network organization in brain networks that have previously been associated with CRCI and MBIs. Additionally, we investigate the impact of MBI on peripheral inflammation. We hypothesized that both MBI and physical training would alter brain connectivity patterns and peripheral inflammation over time compared to no intervention but the effects would be more pronounced after MBI.

## 2. Materials and Methods

The study was registered at ClinicalTrials.gov (NCT03736460). Detailed information about the study protocol has been published elsewhere [51].

### 2.1. Participants

Patients were recruited from the Multidisciplinary Breast Cancer Center, University Hospitals Leuven, after identification through the patient database and via flyers on social media between October 2018 and October 2021. Participants were eligible if they were 18–65 years old, diagnosed with breast cancer with or without solitary metastases (except solitary brain metastases), received chemotherapy and ended this treatment 6–60 months before enrolment in the study and were native Dutch speakers. Participants were excluded in case of MRI contraindications, previous experience with meditation training, or when diagnosed with intellectual disability, neurologic or psychiatric disorders. After screening the medical records, candidates received information via e-mail about the general outline of the study. Interested candidates received the informed consent form and the Cognitive Failure Questionnaire (CFQ) to assess subjective cognitive impairment. For more information about the CFQ, refer to the Appendix A. Only participants with significant cognitive complaints (CFQ total score > 42.9 (mean + 1SD study Ponds et al.) or at least two of the four extra CFQ questions > mean + 1SD study Ponds et al.) were eligible for this study [52]. The study was approved by the Ethics Committee Research UZ/KU Leuven (S59396) and conducted in accordance with the Declaration of Helsinki.

### 2.2. Design and Study Procedure

Participants were randomly assigned to an MBI, a physical training (PT) active control group or a waitlist control (WL) group using an online random number generator MinimPy (http://minimpy.sourceforge.net/, accessed on 2 February 2022) by an independent researcher. Group stratification was based on time since chemotherapy completion, age, and endocrine therapy. Participants in the waitlist control group received no intervention and continued to receive their usual care. No specific instructions were given to this group. Researchers collecting the data were blinded to participants’ group allocation.

Participants in all groups were assessed at three matching time-points: before the intervention (t1), immediately after the intervention (t2), and three months post-intervention (t3). The 3-month follow-up period was chosen based on previous research from our group that showed that this period was sufficient to study the potential long-term effects of MBIs [53,54]. At each time-point, assessments included neuropsychological tests and questionnaires (about one hour), multimodal MRI of the brain (about one hour), and blood sample collection (about 15 min). The neuropsychological [45] and structural [55] findings have been published previously and the task-based MRI findings will be published elsewhere. Participants in both control groups could follow the MBI after finishing all assessments and all participants could withdraw without further follow-up.

### 2.3. Interventions

#### 2.3.1. Mindfulness-Based Intervention

This intervention was based on Mindfulness-Based Stress Reduction [56] and Mindfulness-Based Cognitive Therapy for patients with cancer [57]. The program consisted of four three-hour group sessions spread over eight weeks (one session every two weeks), with in-between online support. The number of in-person group sessions was reduced, because of anticipated dropout, to accommodate participants’ other responsibilities, including jobs, housekeeping, caretaking, etc. [58]. Participants were asked to practice daily at home with audio recordings and they could contact the trainer to ask questions. Each session consisted of guided experiential mindfulness exercises, e.g., focus on the breath, body scan, breathing space, mindful yoga, insight and walking meditation (for more details, see Table 1 in Van der Gucht et al. (2020)), sharing experiences, reflection about experiences during inquiry, psychoeducation, and review of home practices. The program was led by two clinical psychologists, certified as mindfulness trainers with more than ten years of experience in providing MBIs to both clinical and non-clinical populations, including patients with cancer. Attendance in the group sessions was documented by the mindfulness trainer.

#### 2.3.2. Physical Training

This intervention was based on the recommended levels of physical activity for adults [59] and the existing cancer rehabilitation program at the University Hospitals Leuven. The program consisted of four two-hour group sessions spread over eight weeks. Each session consisted of psychoeducation related to physical training, endurance and resistance training, stretching, balance and relaxation exercises, sharing experiences and reviewing homework exercises. Participants were expected to do homework exercises to train endurance for 150 min a week and resistance for 2–3 times a week [59]. The physical training was led by a physiotherapist experienced in oncology rehabilitation. Attendance to group sessions and the amount of home practice was documented by the trainer.

### 2.4. Measures

#### 2.4.1. MRI Acquisition

MRI of the brain was acquired on a 3T Philips Achieva scanner with a 32-channel phased-array head coil. The scanning protocol included 3D-TFE T1-weighted images (voxel size = 0.80 × 0.80 × 0.80 mm^3^, TR/TE = 5.8/2.6ms, and FOV = 256 × 240 × 166 mm^3^) and rsfMRI with a whole-brain T2*-weighted echo-planar imaging (EPI) sequence (voxel size = 1.86 × 1.86 × 2.00 mm^3^, TR/TE = 870/32 ms, and FOV = 208 × 208 × 144 mm^3^, acquisitions time = 6 min, 400 dynamic volumes). During rsfMRI, the lights were turned off and participants were asked to close their eyes, lay still, relax but stay awake. To check whether participants stayed awake, they had to report their level of wakefulness before and after the scan on a scale from 0–10.

#### 2.4.2. Serum Collection and Cytokine Analysis

Venous blood was collected in tubes (two 5 mL BCA) and centrifuged at 1200 g for 10 min. Serum (supernatant) was divided into aliquots and stored at −80 °C until analysis. Inflammatory markers [brain-derived neurotrophic factor (BDNF), monocyte chemoattractant protein 1 (MCP-1), macrophage inflammatory protein (MIP-1β), interferon-α2 (IFN-α2), interferon-β (IFN-β), interferon-γ (IFN-γ), interleukin (IL) 12p70 (IL-12p70), IL-17A, IL-8, IL-18, IL-1β, IL-6, and tumor necrosis factor α (TNFα)] were determined via bead-based multiplex immunoassay (LEGENDPlex, BioLegend, San Diego, CA, USA), with concentration determination via fluorescence-encoded beads on a flow cytometer. Groups were equally distributed across plates, and the mean values across duplicates (separate plate assays) were used for analysis. The coefficient of variation (standard deviation/mean) was calculated to express the extent of variability relative to the mean. The mean intraplate variability was 19% and the mean interplate variability was 33%, suggesting low dispersion of the data relative to the mean.

### 2.5. Image Processing and Analysis

Prior to processing, all images were converted to the Brain Imaging Data Structure (BIDS) [60] and visually inspected for artifacts. Additionally, quantitative quality measures were computed using mriqc 21.0.0rc2 [61]. The data identified as lowest quality, based on visual inspection and quantitative metrics (i.e., signal-to-noise-ratio < mean −1.5 SD on at least one time-point), were checked for exclusion by a neuroradiologist (SS). Preprocessing was performed using fMRIprep 20.2.7 [62], which included realignment, coregistration, susceptibility distortion correction [63], slice-time correction, spatial normalization (MNI152NLin2009cAsym), tissue segmentation, and motion/noise components were calculated (ICA-AROMA) [64]. In the CONN functional connectivity toolbox 21a (implemented in MATLAB R2022a), additional preprocessing was performed, including smoothing (FWHM = 8 mm), linear detrending, and bandpass filtering (0.008–0.09 Hz). Nuisance regression was applied using the following fMRIprep parameters from the fMRIprep output: CSF (5 parameters) and WM (5 parameters) signal, realignment (6 parameters), and ICA-AROMA motion confounds (up to 56 parameters). Nodes of attention-, emotion-, and executive function-related networks (i.e., DMN, DAN, SN, and FPN) were selected from the CONN (v21a) (network) atlas, and the bilateral hippocampi were selected from the Harvard–Oxford subcortical atlas, based on previous research showing the involvement of these regions in both CRCI and MBI. This resulted in a total of 21 regions with 210 connections included in the analysis (see Appendix A). ROI-to-ROI functional connectivity between these regions was estimated using bivariate correlations. Changes in functional connectivity maps over time were compared between all groups. *p*-values were false-discovery-rate (FDR) corrected on ROI level at *p* < 0.05, with an uncorrected connection-level threshold of *p* < 0.01.

### 2.6. Graph Analysis

Graph theory analysis was performed using in-house developed MATLAB (r2022a) scripts and the Brain Connectivity toolbox (v2019-03-03). Weighted connectomes were derived from the absolute correlation values from the ROI-to-ROI analysis with self-connections removed. Weighted graph measures of characteristic path length, global and local efficiency and clustering coefficient were calculated for connections between the nodes described above (see Appendix A). Characteristic path length and global efficiency were calculated using Dijkstra’s algorithm, with the connection–length matrix defined by the inverse edge weights (with self-connections having zero length). Clustering coefficient and local efficiency measures were calculated as recommended by Wang et al. [65]. For each connectome, 1000 random graphs were calculated through random edge permutation, and excluding graphs with disconnected nodes.

Intervention effects were assessed using two-level linear mixed models with a random intercept (participant) and group, time-point, and their interaction as fixed effects in RStudio (version 1.3.1093, lme4) [66]. To account for the non-normality of the residuals, we bootstrapped the *p*-values (boot.pval). Age and time since chemotherapy were tested as covariates, but the covariates were not included in the final models based on backwards selection. All values were scaled so that the standardized coefficients provide information about the effect size [67]. No correction for multiple comparisons was introduced for the number of graph measures due to their intrinsic interdependence.

### 2.7. Serum Cytokine Analysis

A principal component analysis (PCA, RStudio version 1.3.1093) on the correlation matrix of the 13 human inflammatory cytokines/chemokines data was performed to (1) account for the network structure of these markers, especially when involved in neural functioning [68,69], and (2) reduce the number of variables to include in subsequent analyses [70]. Concentrations were log transformed when distributions were non-Gaussian. Based on visual inspection of the scree plot, total variance explained, and eigenvalues > 1, principal components were retained (*n* = 3). Longitudinal group differences in biomarkers of inflammation were estimated using linear mixed models with a random intercept (participant) and with group, time-point, and their interaction as fixed effects. Based on previous research, participant age [69] and days in storage of the blood samples [71] were included as covariates. To identify influential values, Cook’s Distance was computed for all blood markers [72]. All values were scaled so that the standardized coefficients provide information about the effect size [67]. No correction for multiple comparisons was introduced for the number of principal components as they are interdependent.

### 2.8. Correlation Analysis

First, based on the identified significant group-by-time interaction effects, we investigated the correlation between significant changes in graph metrics and changes in the inflammatory profiles. Second, as our primary outcome was the change in cognitive complaints over time [51], we investigated the correlation between changes in CFQ scores and significant changes in graph metrics or inflammatory profiles. Third, we previously identified significant differences between the physical training and waitlist control group on self-report questionnaires assessing emotional well-being (Depression Anxiety Stress Scale; DASS) and fatigue (Checklist Individual Strength; CIS) as part of this larger RCT [45]. Therefore, we additionally calculated the correlation between significant changes on these two self-report questionnaires and significant changes in graph metrics or inflammatory profiles. More details about the questionnaires can be found in the Appendix A. Spearman correlations were used to investigate associations between change scores across all groups. Results were considered significant at *p* < 0.05. Due to the exploratory nature of this analysis, we did not correct for multiple comparisons.

## 3. Results

### 3.1. Enrolment and Attrition

Figure 1 shows detailed information about enrolment and attrition. Of the 657 participants who were assessed for eligibility, 121 breast cancer survivors with cognitive complaints signed the informed consent form. Before the baseline measure, four participants dropped out because of time constraints. Therefore, 117 participants were randomly allocated to the mindfulness (*n* = 43), physical training (*n* = 36), or waitlist control condition (*n* = 38). In total, 95 participants completed the assessments at the three time-points. For the rsfMRI data, we performed a complete case analysis and data of 21 additional participants had to be excluded: 15 participants with signal-to-noise-ratio < mean −1.5 SD on at least one time-point, 4 with failed denoising, and 2 with unspecific neuroanatomical abnormalities. This resulted in the data analysis of 74 participants (30/21/23 MBI/PT/WL). For the blood serum analysis, all available data have been analyzed.

### 3.2. Participant Characteristics

Table 1 describes the demographic and medical information of the participants at baseline. Information on the distribution of chemotherapy regimens can be found in Appendix A. Table 2 shows information regarding home practice. In the total sample (*n* = 117), 69% of the MBI participants practiced at least several times a week at t2, and 51% at t3. Of the physical training participants, 55% practiced at least several times a week at t2, and 35% at t3.

### 3.3. Resting State Functional Connectivity and Graph Analysis

All participants indicated that they stayed awake during the resting state scans. No significant differences in ROI-to-ROI functional connectivity between the treatment groups were observed at baseline, nor between or within groups from pre-to-post intervention (t2–t1 and t3–t1). In terms of functional network organization, at baseline, significant differences in characteristic path length existed, with the MBI and physical training group having higher baseline values than the waitlist control group (mean (SD): MBI = 1.25 (0.07); PT = 1.27 (0.07); WL = 1.21 (0.06); *p* = 0.01). Additionally, the physical training group had significantly lower values of global efficiency at baseline compared to the waitlist control group (mean (SD): PT = 0.87 (0.03); WL = 0.90 (0.02); *p* = 0.01). No other baseline differences between the groups were identified.

Figure 2 and Appendix A show the significant group-by-time interaction effects for the graph measures. Over time, we found a larger decrease in characteristic path length from baseline to post-intervention (β = −0.92, *p* = 0.01) and three-month follow-up (β = −1.20, *p* = 0.03) in the physical training compared to the waitlist control group, with within-group effects showing a significant decrease in characteristic path length in the physical training group (t2–t1: β = −0.70, *p* = 0.01; t3–t1: β = −0.82, *p* = 0.01) and no significant changes in the waitlist group over time. These findings reflect increased functional network integration in the physical training group compared to no intervention.

For the clustering coefficient, a larger decrease from baseline to three-month follow-up was found in the MBI (β = −0.78, *p* = 0.03) and physical training group (β = −0.82, *p* = 0.04) compared to the waitlist group. Within-group effects showed a significant decrease in the MBI (β = −0.65, *p* = 0.01) and physical training group (β = −0.62, *p* = 0.04) at three-month follow-up, and no significant changes in the waitlist group. These findings reflect decreased functional network segregation in both the MBI and physical training group compared to no intervention.

For global efficiency, a larger increase from baseline to post-intervention and three month-follow up was found in the physical training group compared to the waitlist group (t2–t1: β = 1.14, *p* < 0.001; t3–t1: β = 1.05, *p* < 0.001) and MBI (t2–t1: β = 0.70, *p* < 0.001; t3–t1: β = 0.64, *p* < 0.001). Within-group effects showed a significant increase in the physical training group from baseline to post-intervention and at three-month follow-up (t2–t1: β = 0.72, *p* < 0.001; t3–t1: β = 0.72, *p* < 0.001). These findings reflect increased functional network integration in the physical training group compared to MBI and no intervention. Finally, no significant differences in local efficiency were identified.

### 3.4. Serum Cytokine Analysis

The model generated three principal components, which accounted for 64% of the variance (Table 3 and Appendix A). We found a larger increase in principal component 1 (large negative loadings of IL-12p70, IFN-γ, IL-1β, and IL-8) from baseline to immediately post-intervention in the physical training compared to the waitlist group (β = 0.58, *p* = 0.03; Figure 3 and Appendix A). Within-group analysis of principal component 1 was non-significant (Appendix A).

### 3.5. Correlation Analysis

No significant correlations were identified between significant graph metrics or inflammatory profiles, and questionnaires of subjective cognitive impairment, emotional well-being, or fatigue (Appendix A).

## 4. Discussion

In this longitudinal RCT, we investigated the impact of MBI compared to physical training and no intervention on resting state functional connectivity and functional organization in brain networks related to both CRCI and MBI, and on peripheral inflammation in breast cancer survivors with cognitive complaints. No differences in functional connectivity were identified. However, after receiving either MBI or physical training, differences in functional network organization were found compared to no intervention. Additionally, differences in inflammatory profiles were identified after physical training compared to no intervention only.

Contrary to our hypothesis, we could not identify differences between the groups in resting state functional connectivity patterns. Other studies investigating the impact of standardized 6- to 8-week MBIs on functional connectivity did find changes in attention-, emotion-, and executive function-related networks. However, these studies used a variety of seed regions and ROIs depending on the pathology under investigation which makes it difficult to compare the results [33]. In breast cancer survivors with chronic neuropathic pain (*n* = 23), the effects of an 8-week MBI on resting state functional connectivity have been studied using seed-to-whole-brain connectivity analysis with a seed in the DMN (i.e., posterior cingulate cortex). The authors reported increased functional connectivity between the posterior cingulate cortex seed and the medial prefrontal cortex after MBI compared to no intervention, which correlated with a reduction in the experience of pain [34]. In the current study, we focused on neural networks that have previously been linked to both CRCI and MBI. The design was similar to the one of our pilot study, in which we identified functional connectivity differences between the SN and DAN after MBI compared to no intervention in breast cancer survivors with cognitive complaints (*n* = 33) [35]. Hence, the lack of significant findings compared to our pilot study could not be related to differences in the choice of ROIs. Therefore, we speculate that the results of our pilot study might have been driven by a type I error due to the small sample size. Thus, well-powered studies investigating similar ROIs are needed to verify the lack of significant findings. Additionally, although the practice period was comparable with standardized MBIs, the reported amount of home practice might have been insufficient to induce extensive brain changes in the current study.

Alternatively, the lack of significant findings could in general be linked to (1) insufficient power to detect differences, (2) lack of sensitivity due to data quality, and (3) the absence of a true effect. First, studies have detected changes in resting state functional connectivity after MBI compared to a control group in samples with only 10 participants per group [33]. Hence, our study with at least 21 participants per group should be sufficiently powered to detect between-group differences. Second, we combined both qualitative and quantitative data quality control processes to guarantee the data quality of included MR images in this study. Therefore, we believe that insufficient data quality does not underlie the lack of effects. Third, subgroup analyses of the participants that were included for the resting state analysis showed the same results in terms of cognitive complaints, emotional well-being, fatigue, quality of life, and mindfulness skills as for the total sample [45]. Hence, the resting state sample is representative of the total sample, suggesting that the lack of significant findings might indeed be related to a lack of true effects of MBI on resting state functional connectivity in the context of CRCI.

In terms of functional organization of the network encompassing regions related to attention, emotion, and executive functioning, we found reduced segregation as characterized by a lower clustering coefficient after receiving either MBI or physical training compared to no intervention at three-month follow-up. Although this might reflect less cohesion of the neighboring nodes in the network [20], the balance between segregation and integration is essential for efficient cognitive processing and the operation of distributed networks underlying cognitive function. More specifically, when there is less segregation and more integration of the functional network, specialized regions for specific functions might become more interconnected with other brain regions, resulting in a more efficient functional organization of the neural network [73]. As we did not find changes in functional integration in the mindfulness group over time, it is difficult to interpret the segregation results. In the physical training group, however, we found increased integration as characterized by a decrease in characteristic path length and an increase in global efficiency, potentially reflecting more efficient information transfer between nodes in the defined brain networks [21]. Therefore, we speculate that the brain is more efficient in integrating information across the investigated brain network, rather than relying on specialized regions for cognitive processing. This can be beneficial for cognitive tasks that rely on executive functioning, as these higher order cognitive processes depend on the integration of anatomically distributed neurons [74]. When comparing mindfulness to physical training, global efficiency was lower after MBI. More specifically, global efficiency remained stable in the MBI group over time and increased in the physical training group. This, in combination with similar findings for characteristic path length, suggests that MBI does not impact the functional integration of the investigated network. Thus, physical training might have a more pronounced impact on functional network organization of networks related to attention, emotion, and executive functioning.

In a previous study, we did not identify differences between the groups in terms of subjective and objective cognitive impairment after the interventions [45]. Therefore, we speculate that functional network reorganization occurs before behavioral changes, and longer periods of practice might be needed to strengthen the functional network organization and subsequent changes in cognition. Alternatively, it is possible that the differences in objective cognitive impairment were obscured by practice effects, although they might have co-occurred with the functional network changes. Finally, it is possible that the neural network reorganization is not linked to processes related to cognition, but rather to emotion processing.

Finally, although it has been summarized in a systematic review that MBIs might aid in the recovery of the immune system in cancer patients by altering cytokine levels, the evidence remains contradictory [41]. Additionally, the studies included in this review focused on altered levels of individual cytokines, while cytokines are part of an interconnected network. The effect on one cytokine might have little significance in the context of therapeutic interventions to reduce inflammation, as cytokines impact each other to produce a biological effect [75]. Interestingly, cytokines showing differences after chemotherapy when compared to no-chemotherapy or healthy women also loaded heavily on the second inflammatory component (MCP-1, MIP-1β, and BDNF) [69], whilst no intervention effects were found. Thus, higher values of these cytokines might be of interest to differentiate between individuals after chemotherapy treatment, but do not necessarily associate with physical or psychological intervention effects. Another inflammatory component, primarily determined by IL-12p70, IFN-γ, IL-1β, and IL-8, showed significant decreases shortly after physical training when compared to no intervention. In line with these findings, a study on the effects of a 6-week physical training on inflammation in breast cancer patients during chemotherapy treatment showed a reduction in the same pro-inflammatory markers (IFN-γ, IL-1β, and IL-8) from pre-to post-intervention [76]. This indicates that physical training might reduce inflammation [77] in breast cancer survivors, whilst this was not observed after MBI. In turn, reducing therapy-induced immune deregulation might improve cognitive function [25]. In our study, however, we could not find support for an association between self-reported cognitive functioning and inflammation. This might be because we did not identify differences between the groups in cognitive complaints [45], reducing the possibility to find a significant correlation.

### Limitations and Future Research

To better understand the (lack of) treatment effects, future research could assess potential adverse effects using standardized questionnaires, therapist fidelity using the Mindfulness-based Intervention–Teaching Assessment Criteria (MBI-TAC), and homework practice using smartphone monitoring. Additionally, although participants were excluded if they had previous meditation experience and participants in the control groups were expected not to engage in meditation-related practices during the study, the amount of physical exercise and meditation practice could be more specifically assessed in both groups in future research. Furthermore, this study showed that physical training can impact inflammatory profiles compared to no intervention. However, within-group effects were non-significant, suggesting that the group-by-time interaction effect was driven by subtle changes in inflammatory profiles over time. As this study was not designed to test the effects of physical training on CRCI, but merely to use physical training to control for non-specific intervention effects, future studies could adapt the physical training to potentially increase the treatment effect. For example, the physical training could be adapted to individual needs by letting participants select the exercise modality, intensity, and context [78]. This way, dose–response relationships could also be investigated. Furthermore, it would be useful to measure physical activity levels (i.e., VO_2_-max) at baseline and throughout the study, as physical fitness levels might have confounded our effects. Moreover, different types of physical activity can have differential effects on biological outcomes. For example, aerobic exercise might have the largest effect on brain-derived neurotropic factor (BDNF) and anti-inflammatory cytokines, resistance training might have a larger effect on insulin-like growth factor-1, and yoga might have the largest effect on oxidative stress [79]. Tailoring the intervention to target specific biological outcomes might thus be pivotal to increase treatment effects. Hence, our findings in the physical training group should be considered preliminary, and studies incorporating the above-mentioned limitations are needed to provide further insight into the impact of physical training on CRCI. Additionally, as the differences in the inflammatory profiles between the physical training and waitlist group were not maintained at the three-month follow-up, longer interventions might be needed to impact immune deregulation in the long term. Furthermore, we could not find an association between the significant neural and inflammatory findings, and changes in self-report measures. When using change scores, complete cases are needed to correctly calculate the correlations, which limits the power of the correlation analysis. Nonetheless, we already used complete cases for the graph analysis, so we only needed to exclude the participants with missing data from the blood analysis. Additionally, we had to exclude data from 21 participants, mainly due to motion artefacts that lowered the signal-to-noise ratio. To increase the power of the statistical analyses in future studies, the instructions to limit participant motion might be improved. Finally, to provide more general recommendations for the treatment of CRCI, future research should also include survivors with other types of non-central nervous system cancers, sex, gender, ethnicity, and race.

## 5. Conclusions

In this study, we found no evidence for an effect of MBI on resting state functional connectivity in brain networks that have previously been linked to CRCI. However, both MBI and physical training could induce changes in the functional organization of the neural networks related to attention, emotion, and executive functioning. After receiving either MBI or physical training, reduced functional segregation was found, whereas only physical training increased functional integration of the neural network. Hence, physical training might have the most pronounced effect on the efficiency of neural information processing in areas related to attention, emotion and executive functioning. Additionally, physical training might reduce therapy-induced immune deregulation by reducing levels of pro-inflammatory cytokines. This research suggests that both MBI and physical training might impact biomarkers of CRCI in women with cognitive complaints after breast cancer treatment, although the effects were more pronounced after physical training.

## Figures and Tables

**Figure 1 cancers-15-03632-f001:**
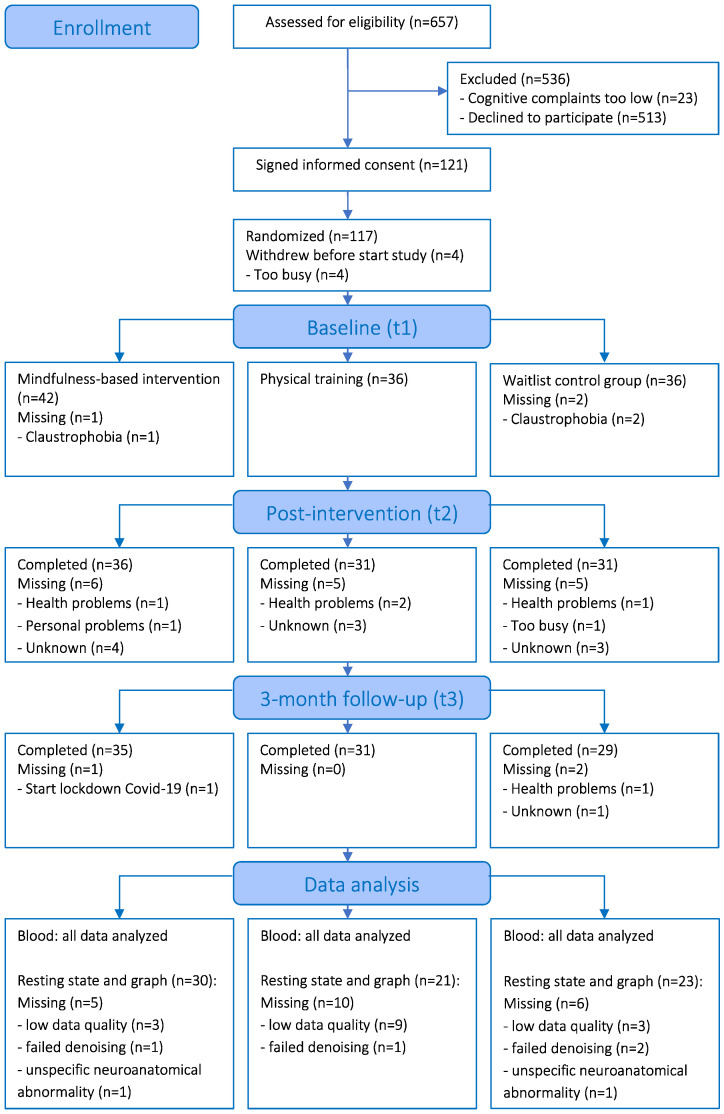
CONSORT flow diagram.

**Figure 2 cancers-15-03632-f002:**
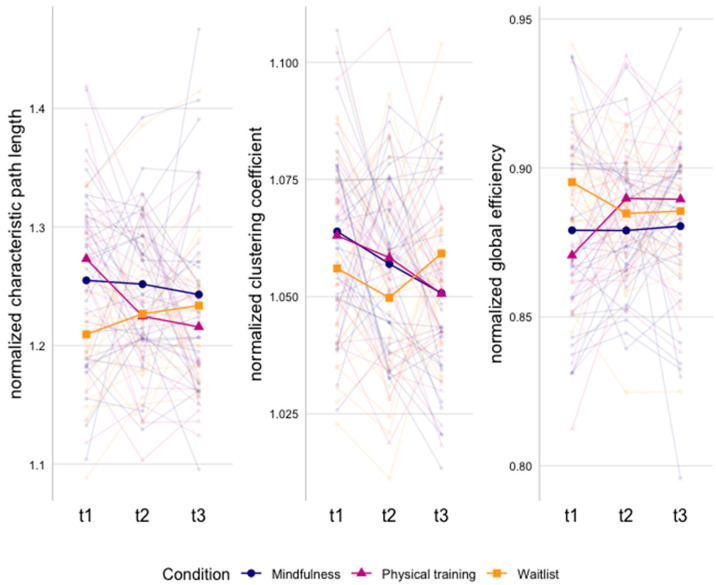
Graph measures showing significant differences between groups over time. Mean values per group and individual lines per subject are presented.

**Figure 3 cancers-15-03632-f003:**
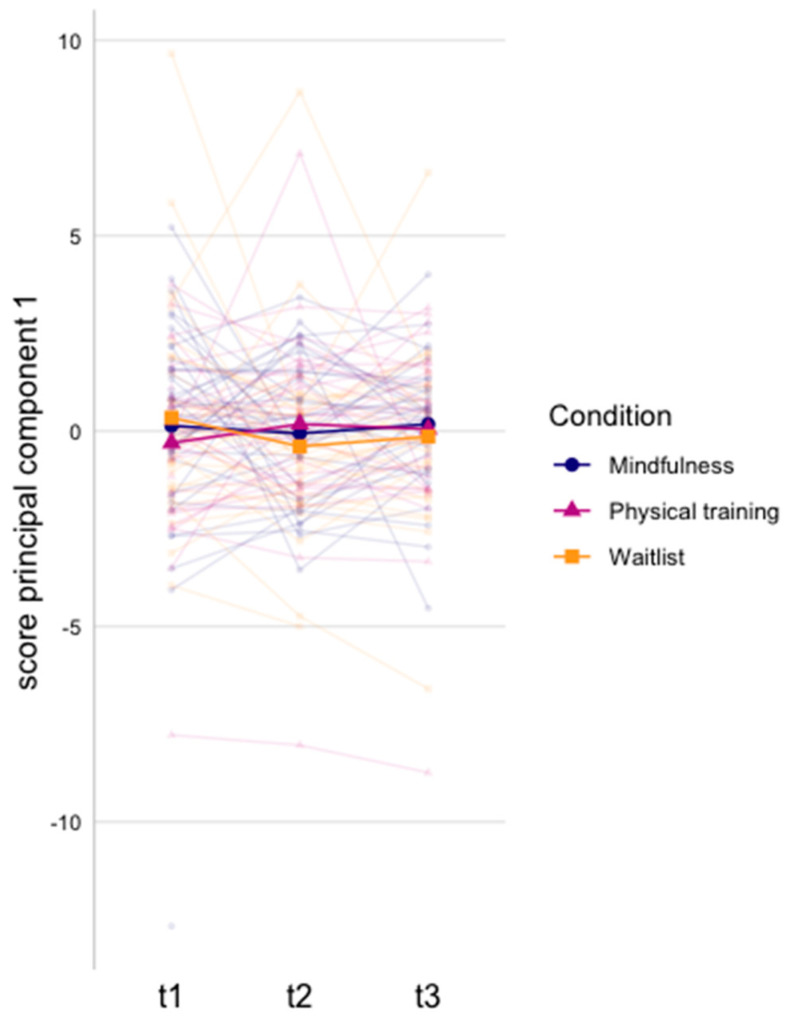
Principal component scores per intervention group over time for principal component 1. Scores were created from concentrations of 13 human inflammatory cytokines/chemokines, of which 3 components were retained. Mean values per group and individual lines per subject are presented.

**Table 1 cancers-15-03632-t001:** Demographic and medical characteristics of participants at baseline.

	Mindfulness		Physical Training		Waitlist	
Mean (SD) [95% CI] or *n* (%)	blood (*n* = 43)	rsfMRI (*n* = 30)	blood (*n* = 36)	rsfMRI (*n* = 21)	blood (*n* = 38)	rsfMRI (*n* = 23)
Age	47.2 (8.1) [44.7, 49.7]	48.1 (7.8) [45.1, 51.2]	48.0 (7.7) [45.4, 50.6]	50.2 (6.9) [42.4, 58.0]	50.1 (10.1) [46.8, 53.4]	50.1 (10.2) [42.4, 57.8]
Time since chemo	24.9 (14.8) [20.4, 29.5]	26.8 (15.8) [21.2, 32.4]	24.5 (13.6) [19.9, 29.1]	25.3 (13.2) [11.0, 39.6]	26.3 (15.1) [21.3, 31.2]	25.5 (16.4) [11.4, 39.5]
Verbal IQ	110 (7.2) [108.0, 112.0]	109.3 (6.2) [107.0, 111.5]	111 (5.3) [109.0, 113.0]	110.2 (6.3) [104.4, 116.1]	107 (5.2) [105.0, 109.0]	108.5 (6.3) [102.8, 114.2]
Chemotherapy	43 (100)	30 (100)	36 (100)	21 (100)	38 (100)	23 (100)
Endocrine therapy	30 (69.8)	22 (73.3)	27 (75.0)	16 (76.2)	26 (68.4)	17 (73.9)
Radiotherapy	27 (62.8)	19 (63.3)	24 (66.7)	16 (76.2)	34 (89.5)	21 (91.3)
Current psychotherapy	10 (23.3)	6 (20.0)	4 (11.1)	5 (23.8)	5 (13.2)	3 (13.0)
Education level						
Secondary school	12 (27.9)	6 (20.0)	11 (30.6)	4 (19.0)	8 (21.1)	6 (26.1)
Higher education	31 (72.1)	24 (80.0)	25 (69.4)	17 (81.0)	30 (78.9)	17 (73.9)
Race: Caucasian	43 (100)	30 (100)	36 (100)	21 (100)	38 (100)	23 (100)

CI = confidence interval; rsfMRI = resting state functional magnetic resonance imaging; SD = standard deviation.

**Table 2 cancers-15-03632-t002:** Amount of home practice reported by participants after the intervention.

Home Practice *n* (%)	Post-Intervention (t2)				Three-Month Follow-Up (t3)			
	Mindfulness		Physical Training		Mindfulness		Physical Training	
	Blood (*n* = 36)	rsfMRI (*n* = 30)	Blood (*n* = 31)	rsfMRI (*n* = 21)	Blood (*n* = 35)	rsfMRI (*n* = 30)	Blood (*n* = 31)	rsfMRI (*n* = 21)
Never	0 (0.0)	0 (0.0)	0 (0.0)	0 (0.0)	0 (0.0)	0 (0.0)	1 (3.2)	1 (4.8)
Less than once a month	0 (0.0)	0 (0.0)	0 (0.0)	0 (0.0)	2 (5.7)	2 (6.7)	0 (0.0)	0 (0.0)
About once a month	0 (0.0)	0 (0.0)	0 (0.0)	0 (0.0)	1 (2.9)	1 (3.3)	0 (0.0)	0 (0.0)
Several times a month	1 (2.8)	1 (3.3)	2 (6.5)	1 (4.8)	4 (11.4)	3 (10.0)	0 (0.0)	0 (0.0)
About once a week	6 (16.7)	5 (16.7)	2 (6.5)	1 (4.8)	8 (22.9)	8 (26.7)	3 (9.7)	3 (14.3)
Several times a week	18 (50.0)	14 (46.7)	17 (54.8)	14 (66.7)	16 (45.7)	14 (46.7)	11 (35.5)	8 (38.1)
Daily	7 (19.4)	7 (23.3)	0 (0.0)	0 (0.0)	2 (5.7)	2 (6.7)	0 (0.0)	0 (0.0)
Not reported	4 (11.1)	3 (10.0)	10 (32.3)	5 (23.8)	2 (5.7)	0 (0.0)	16 (51.6)	9 (42.9)

**Table 3 cancers-15-03632-t003:** Principal component analysis outcomes. Scores were created from concentrations of 13 human inflammatory cytokines/chemokines, of which 3 components were retained. High loadings (>|0.3|) are indicated in bold.

Cytokine	PC1		PC2		PC3	
Loadings (% contribution to the component)						
IL-12p70	−0.37	(13)	−0.12	(1)	−0.27	(7)
IFN-g	−0.35	(12)	−0.18	(3)	−0.05	(0)
IL-1b	−0.34	(12)	−0.20	(4)	−0.09	(1)
IL-8	−0.31	(10)	0.19	(4)	0.03	(0)
MCP-1	−0.16	(3)	0.48	(23)	0.08	(1)
MIP-1b	−0.20	(4)	0.43	(19)	0.29	(9)
BDNF	−0.23	(5)	0.37	(13)	−0.34	(12)
TNF-a	−0.20	(4)	−0.32	(10)	0.07	(1)
IFN-a2	−0.30	(9)	−0.31	(9)	0.00	(0)
IFN-b	−0.30	(9)	0.16	(3)	−0.50	(25)
IL-18	−0.22	(5)	0.22	(5)	0.46	(21)
IL-17A	−0.24	(6)	−0.24	(6)	0.37	(14)
IL-6	−0.29	(8)	−0.03	(0)	0.31	(10)
Model metrics						
Variance (in %)	41		15		9	
Eigenvalue	5.3		1.9		1.1	

BDNF = brain derived neurotrophic factor, IFN = interferon, IL = interleukin, MCP = monocyte chemoattractant protein, MIP = macrophage inflammatory protein; TNF = tumor necrosis factor.

## Data Availability

Data will be provided by the authors upon reasonable request.

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
