# Peer review of "The Impact of Mindfulness on Functional Brain Connectivity and Peripheral Inflammation in Breast Cancer Survivors with Cognitive Complaints"

_cancers, 2023, doi:10.3390/cancers15143632_

Round 1

Reviewer 1 Report

Melis and colleagues investigates the role of mindfulness-based interventions (MBI) relative to physical therapy and a waitlist control group for women with chemotherapy-treated breast cancer experiencing significant cognitive impairments (chemotherapy-related cognitive impairments; CRCI) as part of a larger randomized controlled trial study. Chemotherapy-treated breast cancer survivors with cognitive impairments were randomly placed in either a mindfulness-based intervention, physical therapy, or waitlist control group. Participants were assessed using neuropsychological tests, questionnaires, multimodal MRI, and blood samples at three different timepoints (before and after the intervention as well as 3-months post-intervention). No notable group differences were confirmed for resting state functional connectivity for the MBI groups post-intervention, although significant changes in network efficiency and regulation were observed primarily in the physical activity intervention (although decreased functional network segregation was observed in the MBI group compared to waitlist controls, 3 months post-intervention). In general, the authors’ data demonstrate that the physical activity intervention (active control) provided the most promising effects in protecting against CRCI.

While the main hypothesis related to MBI as an effective intervention against resting state network dysregulation in patients experiencing CRCI was not well supported by the current results, the potential clinical implications of this study are still important, and the clearly defined limitations of the present study highlight the need to build upon these initial results to further investigate the effectiveness of MBI (and exercise) as a potential low-cost, accessible, non-invasive intervention that may hold promise for the treatment of CRCI.

The manuscript is clearly written, and the methodology of this study is well-designed, particularly with the addition of the physical therapy group to improve upon the pilot study. The rationale behind the conclusions within the discussion session were well thought out and supported by previous studies done by this research group as well as other research groups. The limitations of this study are transparent and offer clear avenues for future research. The revisions for the authors to address are mostly minor. 

Major Comments

General comment:

The abstract background and the intro emphasizes the potential effectiveness of the MBI as an intervention against CRCI, compared to physical activity (PA, active control) or passive control (waitlist control). This framing is a bit misleading, as the most robust findings of the study team’s findings demonstrate the robust findings related to the PA active control intervention in increasing functional network integration (path length) in the RS connectivity analyses, and even more enhanced global efficiency in the PA group over both MBI and waitlist control. The authors may consider revising the theoretical framing of the study to emphasize the effectiveness of MBI and PA in protecting against the neural network alterations associated with CRCI. If they choose not to reframe the argument, it should be made clearer why PA is used as a ‘control’ comparison – i.e., because it has already been reliably and routinely demonstrated as an effective intervention? If so, what benefits might MBI have over a PA intervention? The introduction notes that previous, small sample studies investigating an MBI lacked an active control group, which is a limitation the present study aimed to address. I agree that pitting MBI against another effective intervention is a nice comparison for the present study – it would just be important to build up the rationale for why. (i.e., Why is an active control (PA) an ideal comparison group for the MBI intervention? Are common underlying physiological mechanisms altered by both activities? Controlling for the social interaction, environmental enrichment provided by the personal training?)

Specific comments:

Introduction 

-        Line 82 and 119: Include a brief description of graph theory (what is it, what advantages does this method of analysis provide over other methods of analyzing functional brain activity).

-        Line 101, Mindfulness-based interventions are introduced however, there is limited background on what it is, and how it may be unique in comparison to other clinical interventions. It may be important to provide more context on why this particular intervention could potentially alleviate CRCI symptoms (I.e., could we look at other clinical populations and draw hypotheses from them?; https://www.ncbi.nlm.nih.gov/pmc/articles/PMC8882841/) and why they should be investigated (I.e., is it cost effective; https://www.ncbi.nlm.nih.gov/pmc/articles/PMC9425809/)

Materials and Methods

-        Add a brief section describing the waitlist control condition, and any guidelines of restrictions given to this group. For example, were they explicitly told not to engage in meditation or mindfulness activity, or were they told to limit or track their PA levels during the trial to control for potential confounds?

-        Similarly, were any controls placed to ensure the MBI group did not engage in significant amounts of physical therapy, the physical therapy group did not engage in significant amounts of mindfulness-based practices. If so, how was this treated or controlled for (or why is it not a concern)?

-        Line 171, t3 was 3 months post-intervention. It may be beneficial for the reader to understand why 3 months was chosen (I.e., does 3 months constitute enough time to see the long-term effects of physical therapy and/or MBI in healthy or other clinical populations)?

-        Line 179-182, participants had access to in-between online support and instructed to practice daily at home with audio recordings. Were any analyses conducted on the frequency of use of these in-between online supports and audio recordings for each participant?

-        Line 228, please clarify if these coefficients of variations are within the acceptable range. If not, it may be important to explain why results can still be drawn from these analyses (https://www.statology.org/what-is-a-good-coefficient-of-variation/)

Results 

-        Given the range of compliance for the MBI and PA conditions (with between ~70-55% performing the interventions several times weekly at t2, and ~50-35% at t3), how was this treated or controlled for in the FC or graph analyses?

-        Line 385: The correlational analyses between inflammatory profiles and questionnaires of subjective cognitive impairment, emotional well-being, or fatigue, were reported to be all non-significant. For transparency, it would be useful to the reader to present the non-significant stats, if only in a supplemental table. 

Minor Comments: 

-        Line 38, CRCI being linked to inflammatory processes can be more clearly defined (i.e., an increase or decrease in pro- and anti-inflammatory cytokines? activated microglia?, etc.)

-        Line 68-69, the introductory sentence uses talks about breast cancer and then in the next few sentences broadly goes to cancers in general. It may be more applicable to find statistics on the prevalence of CRCI in breast cancer survivors specifically.

-        Line 80-81: As it currently reads, it is not entirely clear why the hippocampus has been specifically introduced here. The authors may consider expanding on the significance of the HPC in CRCI (i.e., part of the DMN/recollection network; a common neurotoxic target of chemotherapy; likely mediator of long-term memory-related cognitive disturbances in breast cancer patients, etc.).

-        Line 148 – consider including specific details related to the chemotherapy treatment regimes of the study sample (i.e., specific drugs, dosages, frequency, treatment duration, perhaps as a supplemental table). 

-        Line 166-170, Clarify if all assessments (neuropsychological tests and questionnaires, multimodal MRI of the brain, and blood sample collections) were all done at the three timepoints. 

-        Line 393: The sentences state “No differences in functional connectivity were identified. However, after MBI and physical training, differences in functional network organization were found compared to no intervention.” This reads as if participants performed both MBI and PA (two interventions). This should be clarified.

-        Line 517: It is also implied that both MBI and PA were conducted within the same intervention “After MBI and physical training, reduced functional segregation was found…”.

Reviewer 2 Report

The investigators present the results of a well-designed study examining the impact of  mindfulness and physical training interventions compared to a waitlist control on functional brain connectivity and markers of inflammation.  The results are very interesting and clinically relevant.  Below are a few comments for the authors to consider:

1) The title is a bit misleading since two active interventions were included.  Did the investigators consider the physical training as a control group?  If yes, this needs some justification since exercise has been used to treat cognitive complaints in cancer survivors and other populations.  This is particularly important since the most robust findings were for the physical training group.

2) On line 168 there is a statement that the neuropsychological test data have been published. They report that the self-report of cognitive function decreased for all groups but there were no group differences. It would also be important to mention that there were also no group differences based on neuropsychological testing. The Discussion should then include potential ways of interpreting the connectivity findings in the absence of differences in self-report or performance- based measures.  For example, the changes in brain function may not be clinically meaningful or they may be clinically important but differences in neuropsychological testing were obscured by practice effects or changes in connectivity occur first but changes in performance and perceived cognitive function occur at a later time.

3) Baseline differences were found for path length; however, I did not see that these differences were controlled in the analysis.  In the Discussion, the investigators mention that regression to the mean is a possible explanation for the results; however, controlling for baseline differences (if possible) would be preferable.

Reviewer 3 Report

This paper describes the impact of mindfulness on functional brain connectivity and peripheral inflammation in breast cancer survivors with cognitive complaints. The findings are potentially informative for the readers of Cancer. But I have several concerns. Major comments 1.    I am concerned about the high dropout rate. Couldn't you have raised the participation rate a little more? It seems that the parts related to people with claustrophobia and the accuracy of images could have been a little more devised. However, it may be difficult because it is a secondary analysis. 2.    It seems that you recruited on the web, but what kind of wording did you use to recruit participants? 3.    The time required for intervention is long, and isn't it a little burdensome to perform in daily clinical practice? Minor comments

1.    The table 3 are cut off in the middle and are difficult to read.

2.    The figure2 & 3 is too small.

3.    Could you describe a little more about CFQ in the method section?

Round 2

Reviewer 1 Report

The authors have addressed all my concerns. The revised manuscript is significantly more clear, and makes an important contribution to the literature.

Reviewer 2 Report

The authors have provided appropriate responses and revisions.

Reviewer 3 Report

The authors have fully responded to the comments. The table has also been modified .